# Relationship between Executive Functions, Mindfulness, Stress, and Performance in Pediatric Emergency Simulations

**DOI:** 10.3390/ijerph17062040

**Published:** 2020-03-19

**Authors:** Kacper Łoś, Jacek Chmielewski, Włodzimierz Łuczyński

**Affiliations:** Department of Medical Simulations, Medical University of Białystok, 15-274 Białystok, Poland

**Keywords:** medical simulation, medical education, mindfulness, executive functions, pediatric emergency, stress

## Abstract

Over the past decade, high-fidelity medical simulation has become an accepted and widely used teaching method in pediatrics. Both simulation and work in the real conditions of emergency departments are accompanied by stress that affects the executive functions of participants. One of the methods for reducing stress among medical students and healthcare professionals is the practice of mindfulness. The aim of this study was to examine whether executive functions, mindfulness, and stress are related to the technical and non-technical skills of medical students participating in medical simulations in pediatrics. The study included 153 final-year medical students. A total of 306 high-fidelity simulations of life-threatening situations involving children were conducted. Results: Stress and the coping mechanism of the participants were correlated to their skills during pediatric simulations. Some components of mindfulness, such as non-judgment and conscious action, were positively related to the skills of medical team leaders. Executive functions correlated with the non-technical skills and mindfulness of the medical students. Conclusions: Stress, mindfulness, and executive functions modeled the behavior and skills of medical students during pediatric simulations of life-threatening events. Further research in this area may prove whether mindfulness training will improve learning outcomes in pediatric emergency medicine.

## 1. Introduction

Pediatric emergency medicine is a demanding field for young doctors and one in which knowledge, experience, and skills are essential. Both procedural and non-technical skills are important. Non-technical skills are also known as teamwork skills and include leadership, teamwork, situational awareness (SA; avoiding fixation error), decision-making, resource management, safe practice, adverse event minimization, and professionalism [1]. Effective and safe teaching methods for these skills are sought, and one of them may be a high-fidelity medical simulation. Over the past decade, simulation has become an accepted and widely used teaching method in pediatrics [2]. Child resuscitation, trauma management, procedural skills, and teamwork (non-technical skills) can be taught safely using this technique [3]. Due to its repeatability and the standardization of assessment methods, a high-fidelity pediatric simulation is also a good tool for scientific research. In particular, simulations can be useful to evaluate the personal characteristics that affect the behavior of the people involved [4].

Both simulation and work in the real-life conditions of emergency departments are accompanied by stress that affects the executive functions (EFs) of the participants, including attention and memory. Attention lapses increase the risk of serious consequences, such as medical errors, failure to recognize life-threatening signs and symptoms, and other essential patient safety issues [5]. Learning through simulation could improve stress management in critical situations. Mindfulness is one of the methods used by healthcare professionals for stress reduction. Mindfulness is the process of intentionally paying attention to each moment with curiosity, openness, and acceptance without judgment [6]. The goal of mindfulness is to empower individuals to respond to situations consciously, rather than automatically. Among medical students, being mindful is associated with decreased stress, anxiety, and depression and with improved mood, self-empathy, and empathy [7]. Therefore, dispositional mindfulness may increase opportunities for young doctors to engage in error detection or correction involving emotional/behavioral regulation and cognitive control because they are more aware of cognition when self-regulatory behavior is needed.

The executive functions may influence the behavior of medical team leaders. The EFs are a collection of processes that are responsible for guiding, directing, and managing cognitive, emotional, and behavioral functions, particularly during novel problem-solving [8]. Specific subdomains that make up this collection of regulatory or management functions include the ability to initiate a behavior, inhibit the competing actions or stimuli, select relevant task goals, plan and organize a means to solve complex problems, shift problem-solving strategies flexibly when necessary, and monitor and evaluate behavior. Impairment of the EFs may cause several challenges, like difficulties dealing with novel situations and forming a reasonable plan that considers the relevant details, the inhibition of inappropriate responses to situations, and increased distractibility [9]. A small group of studies has revealed that higher mindfulness is associated with better executive functioning [10]. 

It is possible that stress, mindfulness, and EFs, and their interrelationships are important players in emergency cases in pediatrics. There is no research published on the subject in the available literature. For this reason, we conducted standardized, repeated high-fidelity pediatric simulations among final-year medical students to assess their mindfulness, EFs, and technical and non-technical skills as leaders of medical teams. We hypothesized that mindfulness, EF, and stress are related to the skills of medical students during simulation activities. Our results might allow us to design strategies to enhance the skills of young doctors to improve the performance of pediatric emergency medicine teams in the future.

## 2. Participants and Methods

### 2.1. Participants

The project was an observational cohort study in a group of final-year medical students (ClinicalTrials.gov ID: NCT03761355). The research was conducted between October 2018 and June 2019 in the Department of Medical Simulations at the Medical University of Białystok, Poland. The inclusion criterion was being a student of medicine and giving consent to participate in the study. The exclusion criterion was pregnancy.

### 2.2. High-fidelity Pediatric Simulations

The simulations were constructed as high-fidelity scenarios of life-threatening situations involving children. The topics included were supraventricular tachycardia, febrile convulsions, bronchial asthma, ketoacidosis, anaphylactic shock, and paracetamol intoxication. The simulations started in the morning and were identical for all groups of students. All groups received the same introduction to the simulator and medical equipment based on checklists and experienced the scenarios in the same order. The task difficulty was intermediate and was evaluated based on the pilot simulations with both students and residents. Each scenario had two equal goals—technical and non-technical. During the simulation, the students played different roles: team manager, medical team member, or actor—the patient’s caregiver. The analysis concerned only the scenarios in which the students acted as team leaders and included an assessment of their skills and interactions with the other participants.

The following data were collected: age, sex, participation in mindfulness training or other secular or religious meditations, medicines taken, and amount of caffeine consumed before the simulation. The number of high-fidelity simulations the students were part of on a given day and during their entire medical studies was also noted. 

Stress and its impact on simulation were assessed both subjectively by the participants (the higher the score, the greater or more discouraging the stress) and by measuring the heart rate and blood pressure before and after the simulation. The stress-coping style of the team leader was diagnosed with the Polish adaptation of the Coping Inventory for Stressful Situations (CISS) questionnaire [11]. The CISS questionnaire consists of 48 statements about different behaviors that are typical for people in distress. Participants have to determine the frequency, on a five-point scale, of a given behavior in stressful, difficult situations. The scores are formatted on three scales: task-oriented style, emotion-oriented style, and avoidant style. 

### 2.3. Procedural and Non-technical Skills

Technical skills were assessed based on checklists designed for each scenario. The assessment was divided into an interview, a physical examination, diagnosis, and treatment. The more points on the scale, the better the technical skills.

Non-technical skills were assessed using the Ottawa Crisis Resource Management Global Rating Scale (Ottawa GRS) and checklist [12,13]. This tool has well-defined rating scales for each of its categories: leadership skills, situational awareness, communication skills, problem-solving, and resource utilization [12]. Each category is measured on a seven-point anchored ordinal scale with descriptive anchors to provide guidelines on alternating points along the scale. For example, for the indirect assessment of situational awareness from “becomes fixated easily despite repeated cues; fails to reassess and re-evaluate situation despite repeated cues; fails to anticipate likely events” = 1 point, to “avoids any fixation error without cues; constantly reassesses and re-evaluates situation without cues; constantly anticipates likely events” = 7 points. The higher the Ottawa GRS score, the better the non-technical skills. Non-technical skills were assessed by two independent instructors/observers during each simulation. Mean scores between the two observers were used as the reference value. 

### 2.4. Mindfulness

Mindfulness was assessed according to the short version of the Five Facet Mindfulness Questionnaire (FFMQ), after Polish adaptation and validation [14,15]. The FFMQ is used to measure the depth of mindfulness, and it evaluates five factors: conscious presence, non-reactivity, non-judgment, observation, and description. The higher the FFMQ score, the higher the level of mindfulness.

### 2.5. Executive Functions

The cognitive functions were assessed with the Behavior Rating Inventory of Executive Functions—Adult (BRIEF-A). It is a 75-item standardized self-report questionnaire constructed with ecological validity in mind to measure executive functioning in daily life situations [16,17]. The results are summarized as a behavioral regulation index (BRI = inhibit, shift, emotional control, monitor, initiate), metacognition index (MI = working memory, plan to organize, organization of materials, task monitor), and global executive composite (GEC = BRI + MI). Lower scores indicate better executive functioning. Values ≥65 are considered abnormally elevated [18].

Data were presented as means and standard deviations (SD) and rates of incidence of a given characteristic in the group of students. Univariate analysis was conducted using the Mann–Whitney U test for continuous variables, and the Chi-square test for the nominal ones. Correlations were performed using Spearman’s rank correlation coefficient. To find the differences between the groups of students with regard to stress-coping style or previous meditation practice, ANOVA and posthoc pair-wise comparisons were performed. *p* < 0.05 was considered statistically significant. Statistical analysis was performed using the Statistica 13 software (StatSoft, Tulsa, Oklahoma, OK, USA). Only students with all data available were included in the analysis.

The study design was approved by the Ethics Committee at the Medical University of Bialystok in accordance with the Declaration of Helsinki (No R-I-002/358/2017). Signed informed consent was obtained from the students. The rate of consent was 85.9%. The main reason for consent refusal was the lack of time to complete the survey. Students who agreed to participate in the study and those who did not give their consent did not differ in sex, age, or in technical and non-technical skill assessment scores.

## 3. Results

The study included 153 medical students, and each of them played the role of team leader twice. Therefore, a total of 306 simulations were carried out. A summary of data on age, sex, caffeine and drug use, previous meditation practices, and results in mindfulness and student EF scales are provided in Table 1.

Praying or previous meditation practice was not correlated with the values obtained on the mindfulness scale (analysis of variance analysis (ANOVA)). The average mindfulness score on the FFQM scale did not differ from that of the reference in the group of young adult Poles [14]. It was also not different from the score obtained in the previous year with a different group of students (data not published).

### 3.1. Technical and Non-Technical Skills 

The average scores for all students in technical and non-technical skills are presented in Table 2. The average score for situational awareness (SA, i.e., avoidance of fixation error) was statistically lower compared to other non-technical skills. A strong positive relationship was noted between procedural and non-technical skills (r = 0.7, *p* < 0.0001).

### 3.2. Stress, Stress-coping Style, and Students’ Skills

The stress-coping style, heart rate, arterial pressure, and subjective assessment of stress related to the simulation are presented in Table 3. We did not note differences in the students’ skills relative to their stress-coping style (ANOVA, *p* > 0.05). In contrast, stress before a simulation was more severe in students with an emotion-oriented stress-coping strategy than in those with a task-oriented strategy (4.3 ± 1.6 vs. 3.3 ± 1.8; *p* = 0.01). Similarly, stress was more discouraging among students with an emotion-oriented coping style than in those with task-oriented and avoidant styles (2.8 ± 0.5, 2.3 ± 0.5, and 2.3 ± 0.4, respectively; *p* < 0.001).

We observed correlations between stress, the role played during the simulation, and the students’ skills. The more mobilizing was the stress perceived by the students, the higher their technical skill score (r = −0.29, *p* = 0.005), and the better their SA (the more often they avoided the fixation error; r = −0.25, *p* < 0.01). If the students played the role of team leader, the stress after the simulation was greater than when they were team members (r = −0.3, *p* < 0.01). The smaller the students’ relative stress during the simulation, the more often they asked for help from a consultant when it was needed (resource utilization skills; r = −0.32, *p* < 0.01). Considering the participants’ previous experience in high-fidelity simulations, the more scenarios a student completed before the start of the study, the more stress felt before performing the task (r = 0.25, *p* = 0.005), and the stress was more discouraging (r = 0.28, *p* < 0.01).

### 3.3. Repeating Simulations in the Same Team

When the same student acted as a team leader in repeated simulations (different scenarios, but with the same level of difficulty), there was a decrease in the perceived stress before and after the next simulation (*p* < 0.001). Furthermore, stress was perceived as more mobilizing in the subsequent simulation (*p* < 0.001).

Regarding the performance in subsequent simulations, the students’ non-technical skills improved (24.3 ± 3.6 vs. 31.9 ± 5.1; *p* < 0.01) but not their procedural skills (6.4 ± 2.0 vs. 7.0 ± 2.2; *p* = 0.1). Similar results were obtained by analyzing the subscales of non-technical skills. The exception was SA, for which the difference was not statistically significant (4.0 ± 1.1 vs. 4.4 ± 1.5; *p* > 0.1).

### 3.4. Mindfulness Components are Related to Students’ Skills During Simulation

The total score on the FFQM scale was not related to any of the student skills. In contrast, non-judgment (one of the components of mindfulness) was positively associated with the total score of technical skills (r = 0.27, *p* < 0.01). In addition, more conscious action was associated with better SA and the use of available forces and resources among non-technical skills (r = 0.28, *p* = 0.001 and r = 0.26, *p* < 0.01, respectively).

### 3.5. Executive Functions and Students’ Skills and Mindfulness

EFs positively correlated with the students’ non-technical actions. The total EF result, i.e., total GEC, was associated with problem-solving skills (r = −0.26, *p* < 0.001). The EFs “plan organize” and “task-monitor” were associated with all non-technical skills (total: r = 0.32, *p* < 0.001), including the strongest association with general impression (r = −0.33, *p* = 0.001), team management (r = −0.30, *p* = 0.001), and team communication (r = −0.29, *p* = 0.001). The EF “organization of materials” correlated with the ability to communicate with the team (r = −0.28, *p* < 0.01), and the summary of the MI correlated with the use of names and closed-loop (r = −0.26, *p* = 0.01).

All EFs of the BRIEF-A tool were also positively correlated with FFMQ mindfulness: BRI (r = −0.49, *p* < 0.0001), MI (r = −0.28, *p* < 0.001), and GEC (r = −0.44, *p* < 0.0001).

## 4. Discussion

Using repeated high-fidelity pediatric simulations, we showed a relationship between features, such as stress, mindfulness, and EFs, and the skills of final-year medical students. These results could be useful in improving the learning outcomes of students and young doctors in the field of pediatric emergency medicine.

We noticed a strong correlation between the technical and non-technical skills of students. A similar relationship has been demonstrated among French Emergency Medical Service workers during simulations at both the individual and team levels [19]. In addition, the performance of simulation participants was positively associated with their self-confidence and negatively associated with their dissatisfaction. Among our students, the perception of stress as mobilizing also correlated with a better result in the assessment of technical skills. Moreover, lower stress correlated with a more frequent request for help from a consultant in situations when it was necessary to complete the task. A novelty in our research was the repetition of simulations in similar conditions, with the same team leader but different tasks. As a result, we observed a decrease in the feeling of stress and stress perceived as more mobilizing by the participants. Similar positive results were obtained in nurses: repeating simulations with the same team reduced stress and anxiety and increased self-confidence in life-threatening situations [20]. However, one should remember that when dealing with, seriously, a child, there will always be a difference in stress between simulated and real cases. The latter will be characterized by a higher emotional impact (“being a parent” vs. “not being a parent”), which can influence both technical and non-technical skills. Either way, simulation instructors should design their scenarios so that their participants learn to control stress and use it for effective learning and action in real cases. 

In our research, we observed lower student performance in terms of situational awareness compared to other non-technical skills. SA was associated with the students’ perception of stress as a mobilizing and conscious component of mindfulness. In contrast to other non-technical skills, SA did not improve in subsequent scenarios. SA, next to communication and leadership, is essential when managing a team in a life-threatening situation [21]. Loss of SA can lead to errors. Occasionally, even though team members may notice something and are able to make a correct diagnosis or suggest proper treatment, they do not speak up. The reasons for this include not wanting to be wrong, not wanting to hurt someone’s feelings, or not being sure [22]. Simulations seem to be a great tool to learn and improve this critical skill in emergency medicine departments [23]. In our scenarios, SA was assessed indirectly by teachers observing the scenario. A direct assessment of this skill could be done with the situation global assessment technique (SAGAT) scale. However, this tool is difficult to use because it requires stopping the simulations several times. In our study, we observed a relationship between SA and the students’ technical skills, including achieving the goals of the scenario. In contrast to our results, similar pediatric simulations have shown no relationship between SA as assessed by the SAGAT method and achievement of the goal of the scenario [24]. In another study, this skill correlated with the team’s clinical performance but did not correlate with the team’s perception of shared understanding, team leader effectiveness, or team experience—similar to our results [25]. There are tools to improve SA. For instance, the simulation-based crisis resource management training implemented among pediatric cardiac intensive care unit providers has improved the reporting of doubts about the appropriate procedure to the team leader [26].

Another method for improving SA may be mindfulness, which plays an important role in the performance of medical teams in the stressful conditions of emergency medicine. In our observations, the total mindfulness score was not related to student skill results. However, non-judgment positively correlated with the total score of technical skills and conscious presence. Certainly, the characteristics of mindfulness and self-compassion in pediatric residents are associated with less stress and greater confidence in compassionate childcare [27]. Among residents working in the intensive care unit, mindfulness is also associated with their performance and communication [28]. On the other hand, it has been observed that lower mindfulness scores among volunteers working as psychosocial emergency care personnel may be related to their primary traumatization (post-traumatic stress disorder, PTSD) [29].

Improving mindfulness is worth considering since it is a modifiable characteristic and can lead to better patient care. It seems that the implementation of mindfulness training in emergency medicine departments is feasible and sustainable [30]. After completing mindfulness courses, residents have admitted increased awareness, self-reflection, self-acceptance at work, and acceptance of their own limitations [31]. Moreover, they have mentioned being more resilient and better at setting priorities and limits. In addition, the residents have asked for help more often and seemed to be more open to feedback. They have also indicated an enhanced sense of compassion for others. In an interesting study, mindfulness meditation training led not only to a change in the perception of stress but also to improvement in technical skills (epinephrine administration, defibrillation) and in teamwork during cardiopulmonary arrest simulation [32].

We noted a strong relationship between EFs and the students’ skills during teamwork. Better performance in EFs was associated with better scores in non-technical skills. To date, no such studies have been conducted, and the literature on the relationship between EFs and medical actions is scarce. From our observations, it should be concluded that EFs correlates with the effectiveness of student teams during pediatric life-threatening situations. It is also known that stress in medical students causes temporary impairment of EFs [33]. However, the evaluation of the EFs of our students was carried out before the simulations. Mindfulness training partially prevents the functional impairment associated with high-stress contexts, such as in a military cohort [34]. Executive impairment is also associated with the risk of professional burnout syndrome among healthcare professionals [35].

In our study, mindfulness features correlated with the EFs of medical students. Mindfulness meditation certainly affects the EFs, but the impact appears to be more specific than general. After mindfulness training, the greatest improvement in EFs is observed in inhibition, while the impact on the updating and shifting domains is variable (reviewed in [36]). Recently, a pilot study showed an improvement in well-being and EFs, including working memory in surgery residents, after an 8-week mindfulness course [37]. The combination of mindfulness meditation and cognitive training can lead to an improvement in decision-making competence [38]. It is possible that characteristics, such as awareness of the inhibitors and facilitators of rationality, development of comprehensive awareness of cognitive and affective biases and how to mitigate them, and engagement of metacognitive processes, such as reflection and mindfulness, will lead to improved effectiveness of medical teams [39]. In the future, medical universities should include mindfulness as part of their medical student training so as to improve the skills of students and young doctors during stressful situations.

The strengths of our study were a large number of participants, the repetition of the simulations, the use of checklists to assess the students’ skills, the use of standardized mindfulness assessment methods, and the unique pediatric simulation scenarios. However, the results of our research should be interpreted with caution. Our study had several limitations: a lack of assessment of the impact of leader behavior on the work of team members; conduction of the study in a simulation center, with its unique procedures and resources; and the indirect assessment of soft skills, including SA. Our results indicated relationships between stress, mindfulness, EFs, and student performance during pediatric simulations. In the future, a standardized, randomized intervention should be performed on a similar group of students to assess the impact of mindfulness training on their EFs and skills during stressful situations when the life of a child is at risk.

## 5. Conclusions

The results of our study indicated that EFs, mindfulness, and stress were related to the skills of medical students during simulations of life-threatening situations involving children. Further randomized studies are needed to examine whether an improvement in mindfulness or EFs increases the effectiveness of team action in emergency medicine.

## Figures and Tables

**Table 1 ijerph-17-02040-t001:** Data on students participating in medical simulations.

Age (years: mean ± SD)	24.5 ± 2.2
Sex (N/%)	
Male	56/36.6%
Female	97/63.4%
Caffeine consumed before simulations (N/%)	
no	58/37.9%
1–3 cups a day	95/62.0%
>3 cups a day	12/7.8%
Taking medicines affecting heart rate (N/%):	
yes	4/2.6%
no	149/97.4%
Meditation/praying (N/%):	
does not practice	55/35.9%
irregularly	57/37.2%
regularly	41/26.8%
Mindfulness in FFMQ scale (mean ± SD)	
conscious presence	3.29 ± 0.5
non-reactivity	2.92 ± 0.7
non-judgment	3.00 ± 0.7
observation	3.42 ± 0.8
description	3.56 ± 0.6
total score in FFQM scale	3.24 ± 0.4
Executive functions in BRIEF-A scale (mean ± SD)	
behavior regulation index (BRI)	63.6 ± 10.8
metacognition index (MI)	60.1 ± 10.0
global executive composite (GEC = BRI + MI)	62.5 ± 9.4
clinically significant decrease in EFs (number and %)	34/22.2%

EFs: executive functions; SD: standard deviation; FFMQ: Five Facet Mindfulness Questionnaire; BRIEF- A: Behavior Rating Inventory of Executive Functions—Adult.

**Table 2 ijerph-17-02040-t002:** Student results in terms of technical and non-technical skills.

Technical Skills (total) *	Mean ± SD 6.8 ± 2.0
Non-technical skills (total) **	28.8 ± 4.8
overall performance	4.7 ± 1.1
leadership skills	4.8 ± 1.1
Problem-solving skills	4.9 ± 1.0
situational awareness skills ***	4.2 ± 1.2
resource utilization skills	4.9 ± 0.9
communication skills	4.9 ± 0.8

* maximum 10 points; ** maximum 42 points; *** mean results for all students were statistically significantly lower than other non-technical skills (*p* < 0.001). SD: standard deviation.

**Table 3 ijerph-17-02040-t003:** Stress-coping style and its perception by students before and after simulations.

Stress-coping style:	N/%
task-oriented style	62/40.5%
avoidant style	37/24.2%
emotion-oriented style	54/35.2%
	Mean ± SD
Mean subjective perception of stress before and after simulation (1—no stress, 10—very stressed)	3.8 ± 1.9 vs. 4.0 ± 2.0 (p > 0.05)
Heart rate before and after the scenario	78.2 ± 10.3 vs. 82.5 ± 17.2 (p > 0.05)
Blood pressure before and after the scenario (systolic/diastolic)	121.3 ± 12.4/77.1 ± 4.8 mmHg vs. 126.2 ± 11.6/80.4 ± 4.8 mmHg (p > 0.05)
Subjective assessment of the influence of stress on the performance during simulation (1—mobilizing, 5—discouraging)	2.44 ± 0.74

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
