# Peer review of "Relationship between Executive Functions, Mindfulness, Stress, and Performance in Pediatric Emergency Simulations"

_ijerph, 2020, doi:10.3390/ijerph17062040_

Round 1

Reviewer 1 Report

Very interesting study. It could have positive repercussion on medical practice and student training.

It is necessary to consider that there is a difference between real or simulated situation: in case of a real child, the emotional impact is much higher and can modify technical and no-technical skills. In real situation one variable to consider is "being a parent" and "not being a parent".

Any future studies will investigate further aspects.

Good job!!

Author Response

Thank You. Yes, we agree that there is a difference between real and simulated situation in the case of the child. We added this comment to the discussion:

„However, one should remember that when dealing with seriously child there will always be a difference in stress between simulated and real cases. The latter will be characterized by a higher emotional impact (“being a parent” vs. “not being a parent”) which can influence both technical and non-technical skills. Either way simulation instructors should design their scenarios so that their participants learned to control stress and use it for effective learning and action in real cases.”

Reviewer 2 Report

This is an interesting study that should lead to more research concerning whether mindfulness training improves learning outcomes.  I have only two comments that I would like the authors to address.

  1. Since most of their work involves correlations, the authors should be careful not to use statements, such as the one in the abstract, that stress and coping mechanism affected skills during simulation (line 17).  Please check the rest of the text as well to be sure cause and effect are not implied.
  2. While high-fidelity simulation is accepted and widely used, it can also be argued that one should use it with some caution.  For example, high-fidelity simulation has been reported to lead to overconfidence in medical students (reference at the link below).  Please incorporate these findings into you paper and, perhaps, explain how further research could help to obviate this potential shortcoming of the use of high-fidelity simulation. 
  3. https://link.springer.com/article/10.1186/s12909-019-1464-7

Author Response

Thank You. Our next study is based on the intervention with the use of mindfulness training in medical simulations.

  1. We have checked whole manuscript and changed the strong statements like affected to the more accurate, scientific language (correlated, related etc). Lines: 17, 68, 150 and 271.
  2. Thank You for Your very interesting comment. However we can not fully agree with this statement. First of all the cited article concerns ALS skills. We are shure that ALS and BLS skills are much better taught with the use of low-fidelity than high-fidelity manekins. It is widely known and used in Polish simulation centers for many years. Second, our scenarios were complicated, high-fidelity simulations with few technical and non-technical goals in each scenario. ALS and BLS were never included in the aims of the scenarios. Third, we found that our students were less stressed in the consecutive simulations but we do not agree that they were overconfident. Students’ self-confidence in simulations depends on many factors, in our opinion the strongest one is the relation between teacher and student during the briefing/debriefing sessions. Summarizing, we respect the results of the cited article but it does not concern our research and we do not fully agree with the conclusions of the authors.

Reviewer 3 Report

The authors have done an excellent job in presenting a high quality observational cohort study. As they also mention in lines: 292-294 "a standardized, randomized intervention should be performed on a similar group of students to assess the impact of mindfulness training on their EFs and skills  during stressful situations"

The authors discuss their findings, fully acknowledging limitations associated with their research design.

Only two minor comments:

  • Line 291: “indirect assessment of soft skills, including SA”

Earlier in section 2.3, the authors explain how non-technical skills (including situational awareness) are assessed: line 111-112 “Non-technical skills were assessed by two independent instructors/observers during each simulation”

Considering that SA relationships are presented in the results sections and further discussed  in the discussion It would be useful at that section (section 2.3) to clarify that this a form of an indirect assessment and give an example of 1 or 2 of the descriptive anchors that the observers used.

  • An appendix presenting a couple of the scenarios used in the simulations that were included in the analysis in a little bit more detail, it will help the reader understand the research procedures involved.

Author Response

Thank You. We added more information with examples concerning the indirect assessment of situational awareness in the section 2.3. In our next project we assess the SA directly with the use of direct „freeze” method.

„For example for the indirect assessment of situational awareness from “becomes fixated easily despite repeated cues; fails to reassess and re-evaluate situation despite repeated cues; fails to anticipate likely events” = 1 point, to „avoids any fixation error without cues; constantly reassesses and re-evaluates situation without cues; constantly anticipates likely events” = 7 points.” The scenarios we use in our simulations are confidential and stored in the central Polish data base of simulation scenarios at the Poznań University of Medical Science: http://www.scenariusze.ump.edu.pl/. These scenarios are only available to the teachers involved in Polish simulation project. You can find an example of the scenario very similar to our in: Addison R, Skinner T, Zhou F, Parsons M. Diabetic Ketoacidosis: An emergency Medicine Simulation Scenario. Cureus. 2017 May 29;9(5):e1286. doi: 10.7759/cureus.1286. https://www.ncbi.nlm.nih.gov/pubmed/28680774